# Targeting Grb2 SH3 Domains with Affimer Proteins Provides Novel Insights into Ras Signalling Modulation

**DOI:** 10.3390/biom14081040

**Published:** 2024-08-22

**Authors:** Anna A. S. Tang, Andrew Macdonald, Michael J. McPherson, Darren C. Tomlinson

**Affiliations:** 1School of Molecular and Cellular Biology, Faculty of Biological Sciences, University of Leeds, Leeds LS2 9JT, UK; a.a.s.tang@leeds.ac.uk (A.A.S.T.); a.macdonald@leeds.ac.uk (A.M.);; 2Astbury Centre for Structural Molecular Biology, School of Chemistry, University of Leeds, Leeds LS2 9JT, UK

**Keywords:** protein domains, protein–protein interactions, cellular signalling

## Abstract

Src homology 3 (SH3) domains play a critical role in mediating protein–protein interactions (PPIs) involved in cell proliferation, migration, and the cytoskeleton. Despite their abundance in the human proteome, the functions and molecular interactions of many SH3 domains remain unknown, and this is in part due to the lack of SH3-domain-specific reagents available for their study. Affimer proteins have been developed as affinity reagents targeting a diverse range of targets, including those involved in PPIs. In this study, Affimer proteins were isolated against both the N- and C-terminal SH3 domains (NSH3 and CSH3) of growth-factor-receptor-bound protein 2 (Grb2), an adapter protein that provides a critical link between cell surface receptors and Ras signalling pathways. Targeting the CSH3 alone for the inhibition of PPIs appeared sufficient for curtailing Ras signalling in mammalian cell lines stimulated with human epidermal growth factor (EGF), which conflicts with the notion that the predominant interactions with Ras activating Son of sevenless (SOS) occur via the NSH3 domain. This result supports a model in which allosteric mechanisms involved in Grb2-SOS1 interaction modulate Ras activation.

## 1. Introduction

Protein domains play a critical role in regulating the specificity of cellular signalling through their ability to recognise and recruit specific targets. Src homology 3 (SH3) domains are some of the most abundant protein domains found in signalling proteins and proteins of the cytoskeleton. Despite low sequence identity, all SH3 domains share the same characteristic structure consisting of approximately 60 amino acids, comprising five anti-parallel beta strands connected by three loops (RT, N-Src, and Distal) and a short 3_10_ helix, packed to form two perpendicular beta sheets (Figure 1) [1,2]. The structure of the SH3 domain can independently fold when expressed in isolation and bind target proteins in vitro [3,4]. Recognizing this led to the simplistic views that SH3 domains can function as independent modules and binding specificities are intrinsic [4,5]. However, a more recent study showed that the binding specificities of SH3 domains cannot simply be transferred from one host protein to another, and in proteins containing multiple SH3 domains, domain shuffling leads to a loss of protein–protein interactions (PPIs) [3].

The growth-factor-receptor-bound protein 2 (Grb2) adapter protein consists of a single Src homology 2 (SH2) domain flanked by two moderately similar SH3 domains (with 32.8% sequence identity and 52.5% similarity) (Figure 2) [6]. It is ubiquitously expressed in all eukaryotic cells and was originally identified as an epidermal growth factor receptor (EGFR)-interacting protein but is now known to interact with a wide variety of proteins [7]. Grb2 plays a pivotal role in linking receptor tyrosine kinases (RTKs, such as EGFR) and nonreceptor tyrosine kinases (e.g., focal adhesion kinase) with the Ras–mitogen-activated protein kinase (MAPK) signalling cascade [7]. In the canonical pathway, the binding of extracellular ligands induces the dimerisation and trans-autophosphorylation of the RTK. Grb2 binds via its SH2 domain, either directly to phosphorylated tyrosine residues on the RTK or via another adapter protein, Shc. The SH3 domains of Grb2 interact with the C-terminal proline-rich (PR) domain of Son of sevenless (SOS), the guanine nucleotide exchange factor (GEF) of Ras [8,9,10,11,12]. It has been suggested that the Grb2-SOS interaction predominantly occurs via the N-terminal SH3 domain (NSH3) of Grb2; however, the binding of the C-terminal SH3 domain (CSH3) may improve affinity and increase the overall stability of the complex [13]. It was also suggested that peptide ligands bind independently to each of the SH2 and SH3 domains [14], and that the binding of the SH2 domain to phosphotyrosine motifs does not alter the affinity of SH3 domains for SOS [15]. However, studies investigating ligand binding to Grb2 SH3 domains show that allosteric mechanisms are involved [16,17,18,19]. Thus, the study of SH3 domain interactions needs to be conducted within the context of its host protein, highlighting the need for SH3 domain inhibitors that function in cells.

Non-antibody protein scaffolds have been validated as alternatives to antibodies, not only serving as investigative tools but also as diagnostic or therapeutic agents. They have also proven useful as tools for studying PPIs. To date, the only scaffold-based binding protein specifically targeting SH3 domains is the monobody scaffold [21,22,23]. With the ability to discriminate between SH3 domains of Src family kinases (SFKs), monobody binding proteins demonstrate high specificity for their respective targets [22,23]. As monobodies can be expressed in mammalian cells, they can potentially be used in cell-based studies. However, this has not been fully explored for monobody binders of SH3 domains, and the scarcity of such examples highlights an ongoing need for their development.

Adhirons (formerly commercialised by Avacta as Affimer proteins) are alternative scaffold-based binding proteins, which were developed as affinity reagents targeting a diverse range of targets, including those involved in PPIs [24,25,26]. The Adhiron scaffold (Figure 3) is based on a consensus sequence of 57 phytocystatins (plant cystatins) in which the ligand-binding sequences within each interaction loop have each been replaced with nine randomized amino acids per loop (excluding cysteine residues) [27]. In the work presented here, we utilised Adhiron/Affimer proteins to specifically target SH3 domains to inhibit their interactions and functions. The Grb2 SH3 domains were chosen as proof-of-concept targets due to their known role in linking activated RTKs to Ras-MAPK signalling via the recruitment of SOS [8,9,10,11,12]. Thus, the ability to impede the Grb2 binding of SOS to restrict downstream signalling events was used to evaluate Affimer efficacy in inhibiting Grb2 SH3 interactions. In this study, the inhibition of the interaction between a SOS-derived peptide and Grb2 was shown using Affimer proteins that specifically targeted the NSH3 domain. Affimer proteins individually targeting the NSH3 and CSH3 domains of Grb2 demonstrated the ability to bind endogenous Grb2 from human cell lines. Furthermore, when expressed intracellularly, Affimer proteins targeting Grb2 SH3 domains appeared to inhibit function, as demonstrated by the modulation of Ras-MAPK signalling. Somewhat surprisingly, downstream phosphorylation of ERK (extracellular signal-regulated kinase) and Akt in human cell lines stimulated with human epidermal growth factor (EGF) was significantly reduced in the presence of Affimer proteins targeting the Grb2 CSH3 domain alone.

## 2. Materials and Methods

### 2.1. Production of BAP-Tagged SH3 Domains

The glutathione S-transferase (GST) sequence in pGEX-4T-1 plasmids encoding SH3 domains was replaced with a biotin acceptor peptide (BAP) sequence for production of N-terminally BAP-tagged SH3 domains. Mid-log cultures (OD600 0.6–0.8) of transformed BL21 Star™ (DE3) *E. coli* cells were induced with 0.1 mM IPTG for 4–5 h at 37 °C and 230 rpm before harvesting and lysing them in a 1:20 culture volume of PBS supplemented with BugBuster^®^ Protein Extraction Reagent (Merck), 1 mg/mL of lysozyme, 1X Halt™ Protease Inhibitor Cocktail (EDTA-free; Thermo Fisher Scientific, Waltham, MA, USA), and 25 units/mL of Benzonase^®^ Nuclease (purity > 99%) (Merck). Lysates were clarified via centrifugation at 12,000× *g* for 30 min, and cleared lysates were stored in aliquots at −80 °C.

### 2.2. Phage Display Biopanning and Phage ELISAs

Phage display was conducted over three biopanning rounds, as previously described [28], with the following specifications. The biotinylated targets and deselection targets were immobilised directly from cleared bacterial cell lysates onto streptavidin- or NeutrAvidin-coated well-plates or magnetic beads. In the first and second biopanning rounds, negative selection was performed against cell lysates from non-transformed BL21 Star™ (DE3) cells. Negative selection against another SH3 domain was only conducted in the final biopanning round. From each screen, 48 randomly picked clones were tested for their ability to bind to the target via phage ELISA, as previously described [28], and positive clones were identified via DNA sequencing.

### 2.3. Affimer Protein Production

Affimer sequences were subcloned into pET11a-derived vectors (with or without a C-terminal cysteine) for production and subsequent purification of protein in BL21 Star™ (DE3) *E. coli* cells, as previously described [24].

### 2.4. Production of GST-Tagged Proteins

pGEX-4T-1 plasmids encoding SH3 domains and full-length Grb2 were gifted to us by Professors John Ladbury and Andrew Macdonald (University of Leeds). GST-tagged proteins were produced in BL21 Star™ (DE3) *E. coli* cells, as described for the BAP-tagged proteins. Cells were harvested and lysed in a 1:20 culture volume of Tris Lysis Buffer (125 mM of Tris-HCl, 150 mM of NaCl, 1 mM of DTT, 1 mM of EDTA, 1% Triton X-100, pH 7.4) supplemented with 1 mg/mL of lysozyme, 1X Halt™ Protease Inhibitor Cocktail (EDTA-free), and 3 U/mL of Benzonase^®^ Nuclease (purity > 99%). Lysates were clarified via centrifugation at 10,000× *g* for 30 min. Protein was purified from the cleared lysate using glutathione affinity chromatography, eluting it in 125 mM of Tris-HCl, 150 mM of NaCl, 1 mM of DTT, 1% Triton X-100, and 50 mM of reduced glutathione at pH 7.4 and then dialysing it into PBS with 10% glycerol.

### 2.5. Pull-Down of Affimer Proteins with GST-Tagged Full-Length Grb2

Twenty-five microliters of Pierce™ Glutathione Magnetic Agarose Beads (Thermo Fisher Scientific) (per pull-down) was incubated with cleared lysate containing GST-Grb2 in a GST Binding/Wash Buffer (125 mM of Tris, 150 mM of NaCl, pH 8.0) supplemented with 1× Casein Blocking Buffer (Sigma-Aldrich) for 1 h at room temp. on a tube rotator. Samples were then transferred to a KingFisher Flex magnetic particle processor (Thermo Fisher Scientific) to wash away any unbound protein and then incubated with Affimer-containing lysates for 1 h. After three washes in GST Binding/Wash Buffer, the beads were released into an SDS reducing buffer (50 mM of Tris-HCl, pH 7; 2% (*w*/*v*) SDS; 0.002% (*w*/*v*) bromophenol blue; 5% (*v*/*v*) glycerol; 5% (*v*/*v*) β-mercaptoethanol) for SDS-PAGE followed by Western blot analysis.

### 2.6. Pull-Down of Endogenous Grb2 with 6xHis-Tagged Affimer Proteins

Procedure was automated on a KingFisher Flex magnetic particle processor (Thermo Fisher Scientific). Twenty-five microliters of Dynabeads^®^ His-Tag Isolation and Pulldown (Invitrogen™) (per pull-down) was incubated with Affimer-containing lysate in a Binding/Wash Buffer (50 mM of sodium phosphate (pH 8.0), 300 mM of NaCl, and 0.01% Tween^®^ 20) supplemented with 1× Casein Blocking Buffer for 10 min. After washing away any unbound protein, the Affimer-labelled beads were incubated for 1 h 30 min with mammalian cell lysates in a Pull-down Buffer (3.25 mM of sodium phosphate (pH 7.4), 70 mM of NaCl, 0.01% Tween^®^ 20). After three washes in Binding/Wash Buffer, the beads were incubated for 10 min in His Elution Buffer (300 mM of Imidazole, 50 mM of sodium phosphate (pH 8.0), 300 mM of NaCl, 0.01% Tween^®^ 20) to release bound protein for SDS-PAGE and Western blot analysis.

### 2.7. SDS-PAGE and Western Blotting Analysis of Pull-Down Assays

Samples were diluted in an SDS reducing buffer (50 mM of Tris-HCl (pH 7), 2% (*w*/*v*) SDS, 0.002% (*w*/*v*) bromophenol blue, 5% (*v*/*v*) glycerol, 5% (*v*/*v*) β-mercaptoethanol) and denatured at 95 °C for 5 min prior to separation via SDS-PAGE. Protein blot transfer onto nitrocellulose membrane was achieved using the Trans-Blot^®^ Turbo™ Transfer System (Bio-Rad). Membranes were blocked with 5% (*w*/*v*) non-fat dry milk in tris-buffered saline (50 mM of Tris, 150 mM of NaCl, pH 7.4) with 0.1% Tween^®^ 20 (TBST) for 1 h at room temp and then incubated overnight at 4 °C with antibody diluted in 5% milk-TBST (manufacturer’s recommended dilutions). After washing it 3 times in TBST, the blot was incubated with HRP-conjugated secondary antibody for 1 h at room temp., or HRP-conjugated primary antibody was detected with Immobilon^®^ Forte Western HRP Substrate (Merck Millipore) and imaged.

Antibodies: HRP anti-6X His tag^®^ antibody (Abcam, Cambridge, UK); HRP anti-GST antibody (Abcam); anti-GRB2 antibody (Abcam); anti-rabbit IgG, HRP-linked Antibody (Cell Signaling Technology, Danvers, MA, USA).

### 2.8. Sulfhydryl Biotinylation of Affimer Protein

Purified Affimer proteins with C-terminal cysteines were reduced with Pierce™ Immobilized TCEP Disulfide Reducing Gel (Thermo Fisher Scientific). EZ-Link™ BMCC-Biotin (Thermo Fisher Scientific) was prepared at a concentration of 8 mM in DMSO and added to the reduced protein in a 10-fold molar excess amount. After an overnight incubation at 4 °C, excess nonreacted biotin was removed via size exclusion using Zeba Spin Desalting Columns, 7K MWCO (Thermo Fisher Scientific), before mixing with an equal volume of 80% glycerol for storage at −20 °C.

### 2.9. ELISA for Testing Affimer Cross-Reactivity

Pierce™ Streptavidin High-Binding-Capacity Coated Plates (Thermo Fisher Scientific) were blocked (overnight at 37 °C) with 2× Casein Blocking Buffer in PBS with 0.1% Tween^®^ 20 (PBST). After being washed three times in PBST, 1 μg of biotinylated Affimer in 2× Casein-PBST was added and incubated for 1 h at room temp. After being washed three times in PBST, GST-tagged SH3 domain in 2× Casein-PBST was added and incubated for 1 h at room temp. After washing it three times in PBST, HRP Anti-GST antibody (Abcam) diluted 1:10,000 in 2× Casein-PBST was added and incubated for 1 h at room temp. After washing it six times in PBST, TMB substrate for HRP (SeramunBlau^®^ fast, Seramun Diagnostica GmbH, Heidesee, Germany) was added and incubated for 2 min before measuring absorbance at 620 nm.

### 2.10. Production of Full-Length Recombinant Grb2

A pET28a-derived plasmid containing His-TEV-Grb2 sequence was gifted to us by Professor John Ladbury (University of Leeds). 6xHis-tagged Grb2 was produced in BL21 Star™ (DE3) *E. coli* cells, as described for BAP-tagged SH3 domains (Section 2.1); however, antibiotic selection was executed with 50 μg/mL kanamycin. Cells were harvested and lysed, and protein was purified, as previously described for Affimer proteins (Section 2.3).

### 2.11. Fluorescence Polarisation (FP) Competition Assay

The FITC-labelled SOS-derived peptide (FITC-(Ahx)-VPPPVPPRRR-NH2) was provided by Professor Andy Wilson (University of Leeds). Assays were set up in 96-well, black, non-binding microplates (Greiner Bio-One). Wells were blocked with Casein Blocking Buffer 10× (Sigma-Aldrich, St. Louis, MO, USA) for 3 h at 37 °C and then washed three times with FP Buffer (50 mM of Tris-HCl, 100 mM of NaCl, pH 7.4). A total of 32 μM of Affimer protein and 8 μM of full-length Grb2 was preincubated for 30 min before adding 20 nM of FITC-peptide in a total reaction volume of 100 μL FP Buffer. Fluorescence polarisation (FP) was measured using a Tecan Spark™ Multimode Microplate Reader. FP values were normalised against FITC-peptide only and converted to percentage bound to Grb2 compared with the “No Affimer” control. Datasets from three independent experiments (*n* = 3) were plotted as column bar graphs using GraphPad Prism software (version 10.1.0). A one-way ANOVA with Dunnett’s multiple comparisons test was applied, comparing the mean of each Affimer with the mean of the “No Affimer” control.

### 2.12. Mammalian Cell Culture

U-2 OS human osteosarcoma cells (ATCC^®^ HTB-96™), HEK-293 human embryonic kidney cells (ATCC^®^ CRL-1573™), and 293T human embryonic kidney cells containing the SV40 T-antigen (ATCC^®^ CRL-3216™) were authenticated using short tandem repeat (STR) profiling and were mycoplasma-free. All cell lines were maintained at 37 °C with 5% carbon dioxide (CO_2_) in Dulbecco′s Modified Eagle′s Medium (DMEM) supplemented with 10% (*v*/*v*) fetal bovine serum (FBS) and 1% (*v*/*v*) penicillin–streptomycin. Cell lines generated for stable expression of Affimer proteins were maintained in DMEM supplemented with 10% (*v*/*v*) FBS, 1% (*v*/*v*) penicillin–streptomycin, and 1 μg/mL of puromycin. Affimer gene expression in cells generated with a tetracycline-responsive element (TRE) promoter (Tet-On system) was induced by adding 2 μg/mL of doxycycline (dox) to the cell culture medium.

### 2.13. Mammalian Cell Transfection

Affimer sequences were subcloned between the *Sgf*I and *Mlu*I restriction sites of the pCMV6-AC-GFP vector (OriGene Technologies). HEK-293 cells were seeded in poly-D-lysine-coated 96-well culture plates at 1 × 10^4^ cells per well and incubated overnight at 37 °C with 5% CO_2_ before transfection at ~70% confluency. Transfection was achieved with Lipofectamine™ 2000 Transfection Reagent (Invitrogen™). Cells were incubated at 37 °C with 5% CO_2_ for 24–48 h, and GFP expression was observed using a FITC filter on a Nikon ECLIPSE Ts2R-FL inverted microscope.

### 2.14. Measuring Nuclear Phospho-ERK (p-ERK) in Transfected Cells

Forty-eight hours post-transfection, cells were serum-starved for 1 h before stimulating them with 25 ng/mL of human epidermal growth factor (EGF) for 5 min, rinsing them with Dulbecco’s phosphate-buffered saline (DPBS), fixing them with 4% paraformaldehyde (VWR) for 15 min, and rinsing them again with DPBS. Fixed cells were permeabilised with methanol at −20 °C for 10 min, washed with PBS, blocked in 5% (*w*/*v*) BSA and 0.3% (*v*/*v*) Triton X-100 in PBS for 1 h at room temp, and then incubated overnight at 4 °C with p-ERK antibody (Cell Signaling Technology) diluted 1:100 in blocking solution. After being washed three times in PBS, cells were incubated with anti-rabbit Alexa Fluor™ 633 and Hoechst 33342, both diluted 1:1000 in 1% (*w*/*v*) non-fat dry milk in PBS, for 1 h at room temp, followed by a final set of three washes. Cell images were captured using the ImageXpress^®^ Pico Automated Cell Imaging System and analysed using CellReporterXpress 2.8 (Molecular Devices, San Jose, CA, USA). A cutoff of 750 fluorescence units in FITC nuclear intensity was used to identify Affimer-tGFP-positive cells, and background nuclear p-ERK fluorescence from unstimulated Ala control cells was subtracted. Nuclear p-ERK levels in Affimer-tGFP-positive cells, relative to Ala control-expressing cells, were plotted as column bar graphs using GraphPad Prism software. Datasets representing mean and standard error of mean (SEM) were obtained from three independent experiments (*n* = 3), and statistical significance was determined using one-way ANOVA with Dunnett’s multiple comparisons tests.

### 2.15. Generating U-2 OS Cells Stably Expressing tGFP-Tagged Affimer

The tGFP-tagged Ala control Affimer sequence was subcloned between the *Bam*HI and *Eco*RV restriction sites of the Gateway™ pENTR™ 4 Dual Selection Vector (Invitrogen™). Gateway recombination cloning was then used to further subclone the Ala Affimer-tGFP sequence into the pCW57.1 lentiviral transfer plasmid (Addgene plasmid #41393, a gift from David Root). Other Affimer sequences were then directly subcloned between the *Sgf*I and *Mlu*I restriction sites of the newly generated pCW57.1_Ala plasmid. The pCW57.1_Affimer plasmids were co-transfected with pCMV-VSV-G envelope plasmid (Addgene plasmid #8454, a gift from Bob Weinberg) and pCMV-dR8.2 dvpr packaging plasmid (Addgene plasmid #8455, a gift from Bob Weinberg) into 293T cells to generate lentiviral particles for transduction of target U-2 OS cells [29], as previously described by Tandon et al. [29]. Fluorescence-activated cell sorting (FACS) was used to sort doxycycline-induced, Affimer-tGFP-positive cells, and this was carried out by Dr Ruth Hughes (Bio-imaging and Flow Cytometry Facility, University of Leeds) using a BD FACSMelody™ Cell Sorter.

### 2.16. Epidermal Growth Factor Receptor (EGFR) Signalling Assays in U-2 OS Cells Stably Expressing tGFP-Tagged Affimer

Cells were seeded at 3 × 10^5^ cells per well in 6-well plates and cultured for 48 h before being serum-starved for 1 h, stimulated with 25 ng/mL of human EGF for 5 min, and rinsed with ice-cold DPBS. Cells were lysed on ice for 20 min with NP-40 Lysis Buffer (150 mM of NaCl, 1.0% NP-40, 50 mM of Tris pH 8.0, 1× Halt Protease Inhibitor Cocktail (EDTA-free) and 1× Phosphatase Inhibitor Cocktail 2 (Sigma-Aldrich)), clarified via centrifugation at 16,000× *g* for 10 min at 4 °C, and stored at −80 °C prior to determination of protein concentration using a BCA assay. Twenty micrograms of total protein was analysed using SDS-PAGE and Western blotting (see method 2.7) for detection of α/β-tubulin, Grb2, total ERK, and phospho-ERK. Membranes were then stripped (2× 10 min incubations in 0.2 M glycine (pH 2.2), 0.1% (*v*/*v*) SDS, 1% (*v*/*v*) Tween^®^20) and re-blocked for detection of tGFP-tagged Affimer, total Akt, and phospho-Akt. Western blot protein bands were quantified via densitometry using ImageQuant™ TL software v8.1 (GE Healthcare Life Sciences, Marlborough, MA, USA). Signal intensities were normalised to the tubulin loading control. Datasets from three independent experiments (*n* = 3) were plotted as column bar graphs using GraphPad Prism software. Statistical significance was determined using one-way ANOVA along with Dunnett’s multiple comparisons tests.

Antibodies: α/β-tubulin antibody (Cell Signaling Technology); anti-GRB2 antibody (Abcam); anti-ERK1 + ERK2 antibody (Abcam); anti-Erk1 (pT202/pY204) + Erk2 (pT185/pY187) antibody (Abcam); Mouse monoclonal turboGFP antibody (OriGene Technologies); Akt (pan) (40D4) Mouse mAb (Cell Signaling Technology); Phospho-Akt (Thr308) (D25E6) XP Rabbit mAb (Cell Signaling Technology); anti-rabbit IgG, HRP-linked antibody (Cell Signaling Technology); and anti-mouse IgG, HRP-linked antibody (Cell Signaling Technology).

## 3. Results

### 3.1. Isolation of Affimer Proteins That Bind to the SH3 Domains of Grb2

The initial aim was to isolate Affimer proteins targeting the two SH3 domains of Grb2 through phage display. Proteins tagged with a biotin acceptor peptide (BAP) sequence (GLNDIFEAQKIEWHE) are enzymatically biotinylated in vivo when expressed in *E. coli* and can be immobilised onto streptavidin- or NeutrAvidin-coated surfaces directly from cell lysates. BAP-tagged Grb2 NSH3 and CSH3 domains were presented as targets for three rounds of biopanning against Affimer phage display libraries. In the final round of panning, phages were negatively selected against BAP-Grb2 CSH3 for selection against the NSH3 domain, and vice versa, to remove any cross-reactive binders. From each screen, 48 phage clones were randomly selected to test for their ability to bind to the target via phage ELISA. Of the 48 phage clones selected against BAP-Grb2 NSH3, just one (N-D8) failed to bind the intended target, and little or no cross-reactive binding to BAP-Grb2 CSH3 was observed (Figure 4a). From those selected against BAP-Grb2 CSH3, 38 phage clones bound to the target without binding the negative wells or the BAP-Grb2 NSH3 domain (Figure 4b). The selected binders were sequenced, revealing 17 unique sequences from the BAP-Grb2 NSH3 domain screen and 12 from the BAP-Grb2 CSH3 domain screen (Table 1). Only the variable regions (VR1 and VR2) are shown. Binders with an AAE sequence in VR2 were isolated from an Affimer library that has randomised residues in VR1 only (referred to as the single-loop library). Interestingly, the isolated reagents showed a high proportion of proline residues (highlighted in bold), with some containing known consensus SH3 binding motifs (underlined). The VR1 amino acid sequences obtained for the Grb2 CSH3 screen were particularly interesting, as 23/48 (47.9%) of those sequenced shared a hydrophobic consensus sequence of V/I-M/Q/K-R-P-W/Y/F-W/Y/F-x-S-S.

### 3.2. Affimer Proteins Bound Full-Length Recombinant and Endogenous Grb2 in Pull-Down Assays

All 29 unique Affimer sequences were expressed and purified. The ability to bind full-length Grb2 was initially tested via pull-down assays with glutathione-immobilised GST-tagged Grb2, analysed using SDS-PAGE and Western blotting (Figure 5a–c). Interestingly, the single-loop Grb2 CSH3 binders (C-A8, C-D9, C-F11, C-C10, C-B10, C-A11) did not bind as well as C-D11, which had the same VR1-binding consensus (Figure 5a). This possibly indicates that both VR1 and VR2 loop regions are involved in the binding interaction. Also, two of the binders (C-F8 and C-D10) that failed to show binding in the phage ELISA (Figure 4b) were efficiently able to bind to GST-Grb2 in the pull-down assay (Figure 5a,b). This might be explained by the poor propagation of the phage used in the ELISAs.

Based on pull-down efficiency, sequences, and the frequency of isolation, a selection of these Affimer proteins was further tested for the ability to pull-down endogenous Grb2 from mammalian cell lines. The Western blots shown in Figure 6 are representative of pull-downs from input cell lysates from U-2 OS cells. Pull-downs from HEK-293 cell lysates yielded similar results (Appendix A). Some of the selected Affimer proteins did not bind as efficiently to endogenous Grb2 via pull-down compared to recombinantly expressed Grb2, which could be due to protein conformation and/or the availability of putative binding sites.

### 3.3. Affimer Proteins Bound Exclusively to Their Intended Target in an ELISA-Based Cross-Reactivity Assay

An ELISA-based strategy for testing the cross-reactivity of Grb2-specific Affimer proteins is illustrated in Figure 7a. Biotinylated Affimer proteins were immobilised in streptavidin-coated well plates to incubate with a panel of eleven different GST-tagged SH3 domains and GST as a negative control. Interacting proteins were detected with an anti-GST (HRP) antibody. The Grb2 NSH3 binders (N-D7, N-A9, N-C9, N-B10, and N-A11) were able to capture GST-Grb2 NSH3 with no cross-reactive binding of the other GST-SH3 domains nor with GST alone (Figure 7b). The Grb2 CSH3 binders (C-D10, C-C12, and C-G11) were able to bind GST-Grb2 CSH3, also with no cross-reactive binding of the other tested proteins.

### 3.4. Affimer Proteins Isolated against Grb2 NSH3 Can Inhibit the Interaction between a SOS-Derived Peptide and Grb2

Grb2 is capable of binding SOS-derived peptides, such as the ac-VPPPVPPRRR-nh2 peptide shown bound to the Grb2 NSH3 domain in Figure 1 [2,30,31,32]. Potentially, this peptide could be used to investigate whether its interaction with Grb2 can be inhibited by any of the Grb2 SH3 Affimer proteins. Assays based on the technique of fluorescence polarisation (FP) have been successfully used to screen for inhibitors of PPIs [33]. The basic principle of an FP-based assay is described in Figure 8a. When a small fluorescent peptide is excited with plane-polarised light, it rotates quickly in solution, emitting depolarised light. When bound to a larger molecule, tumbling is slowed, and more of the emitted light retains its polarisation. In the presence of an inhibitor that binds the protein, the fluorescent peptide is released and tumbles quickly upon excitation. A FITC-labelled SOS-derived peptide was utilised in a competition assay to test whether any of the Grb2 SH3 Affimer proteins could inhibit the interaction between Grb2 and a fluorophore-labelled SOS-derived peptide (Figure 8b). All the Grb2 NSH3 binders tested in this assay (N-D7, N-A9, N-C9, N-B10, and N-A11) significantly reduced the binding of the FITC-labelled SOS-derived peptide with full-length Grb2. Affimer proteins targeting the Grb2 CSH3 domain (C-C12, C-G11, and C-D10) were not expected to bind the same region of full-length Grb2 as the peptide and did not reduce binding, as expected.

### 3.5. Affimer Protein Targeting the Grb2 CSH3 Alone Appears Sufficient for Curtailing Ras Signalling in Epidermal-Growth-Factor-Stimulated HEK-293 and U-2 OS Cells

Ras-binding Affimer proteins have previously been shown to bind intracellular Ras and inhibit downstream signalling in HEK-293 cells [25]. We used a similar approach to determine if Grb2-binding Affimer proteins could also inhibit downstream Ras-MAPK signalling (Figure 9). HEK-293 cells were transiently transfected with plasmids encoding Affimer proteins tagged with turboGFP (tGFP), which enabled visualisation of Affimer-producing cells. The cells were then serum-starved and stimulated with human epidermal growth factor (EGF), which usually leads to the downstream phosphorylation of ERK1 via MAPK signalling. Nuclear phosphor-ERK (p-ERK) levels were quantified in Affimer-producing cells only.

Ras-binding Affimer K6 was tested as a positive control, reducing nuclear p-ERK levels by ~68% (*p* ≤ 0.0001) when compared with cells producing a non-protein-binding Ala control (an Affimer that displays four Alanine residues in VR1 and AAE residues in VR2). We tested three of our Grb2-binding Affimer proteins (N-D7, C-C12, and C-D11) and used these to investigate the impact on blocking one or both (via co-transfection) of the Grb2 SH3 domains. Affimer N-D7 alone did not significantly reduce p-ERK levels, but surprisingly, Affimer C-C12 alone reduced p-ERK by ~46% (*p* ≤ 0.01) relative to the Ala control. This was unexpected, as the binding of SOS by Grb2 was thought to be dominated by interactions with the NSH3 domain [13], and, therefore, targeting CSH3 alone for inhibition of PPIs was not expected to curtail the binding of SOS. This reduction in p-ERK in C-C12-producing cells was further enhanced in cells co-transfected with plasmid encoding N-D7, reducing p-ERK levels by ~70% (*p* ≤ 0.0001), equivalent to the direct inhibition of Ras by K6. Another CSH3-targeting Affimer, C-D11, only reduced p-ERK levels when combined with N-D7. However, C-D11 showed much weaker binding to Grb2 in pull-down assays compared with C-C12 (Figure 5 and Figure 6).

To examine global cellular signalling, C-C12 and N-D7 tagged with turboGFP were stably expressed in U-2 OS cells under the effect of a doxycycline-inducible promoter. In the cells expressing Affimer C-C12, p-ERK levels were reduced by ~56% (*p* ≤ 0.05), and the levels of phosphorylated Akt (phosphor-Akt or p-Akt) were reduced by ~83.5% (*p* ≤ 0.05) (Figure 10b–d). As before, Affimer N-D7 did not reduce downstream signalling to significant levels. However, both Grb2-binding Affimer proteins (N-D7 and C-C12) resulted in increased levels of Grb2 (Figure 10e), suggesting the SH3 domains are involved in regulating Grb2 protein stability. Further investigations are required to determine the downstream consequences or effects of this. Nevertheless, this set of experiments confirmed that targeting the Grb2 CSH3 alone for PPI inhibition appears sufficient for curtailing downstream signalling in EGF-stimulated cells and protein stability.

## 4. Discussion

The purpose of this study was to determine if Affimer proteins could function as inhibitors of PPIs with SH3 domains. Affimer N-D7, which targeted the Grb2 NSH3 domain, demonstrated inhibition of a SOS-derived peptide interacting with Grb2, but it only appeared to restrict downstream cell-signalling events in the presence of additional Affimer proteins targeting the Grb2 CSH3 domain. In contrast, targeting the Grb2 CSH3 domain alone with Affimer C-C12 was sufficient for reducing the phosphorylation of ERK and Akt to significant levels, and this was further enhanced with the concerted effect of Affimer N-D7 targeting the NSH3 domain. This result conflicts with the generally accepted view that the interaction of SOS with Grb2 is mediated primarily via the Grb2 NSH3 domain and interactions with the CSH3 domain are not necessary for complex formation but may have an additive effect on binding affinity [13]. If both N-D7 and C-C12 block Grb2-SOS interactions in cells, then this result suggest the opposite is true and the CSH3 domain is primarily responsible for complex formation. Alternatively, it suggests that whilst the CSH3 domain is not necessary for complex formation, it is necessary for the propagation of downstream signalling events.

Evidence dissecting the allosteric mechanisms involved in Grb2-SOS1 interaction led to the previously proposed model depicted in Figure 11, which supports the claim that the CSH3 domain is required for the activation of Ras by SOS1 [18,19]. In this proposed model of Grb2-SOS1 interaction, cytosolic Grb2 is constitutively bound via the NSH3 domain to proline-rich motif 3 (PRM 3) in SOS1, but autoinhibitory mechanisms prevent the CSH3 domain from binding SOS1 PRM 4. In stimulated cells, the Grb2-SOS1 complex is recruited to the plasma membrane, where the Grb2 SH2 domain binds phosphorylated tyrosine residues on the activated RTK. This interaction enables the Grb2 SH2 domain to bind phospholipids in the plasma membrane, inducing conformational changes in Grb2, which then allows the CSH3 domain to bind SOS1 PRM 4 to induce activation of Ras. Based on this model, blocking the CSH3 domain from binding SOS1 PRM 4 should inhibit SOS1-based activation of Ras. Therefore, our data suggest that Affimer C-C12 inhibits this interaction. The binding of Affimer N-D7 to the Grb2 NSH3 domain should prevent interaction with SOS1 PRM3 and subsequent translocation to activated RTKs. However, this would require Affimer N-D7 to be present in a sufficient quantity and have sufficient affinity to outcompete Grb2-SOS1 interaction. Our results indicate that Affimer N-D7 was unable to prevent SOS translocation and subsequent Ras activation to a significant degree.

The proposed model of C-C12’s function assumes it inhibits Grb2’s interactions with SOS. The Grb2 CSH3 domain also interacts with the Grb2-associated binder (Gab) docking proteins Gab 1 and Gab2 [34,35]. The recruitment of SOS and Gab protein occurs via distinct non-overlapping binding sites, and steric hindrance and allosteric conformational changes prevent the formation of a SOS-Grb2-Gab ternary complex [16]. The recruitment of Gab1/2 to activated EGFR via Grb2 provides docking sites for SH2 domain-containing protein tyrosine phosphatase 2 (SHP2) and the p85 subunit of phosphoinositide 3-kinase (PI3K) [36,37]. SHP2 promotes Ras activation by dephosphorylating RasGAP (Ras GTPase-activating proteins) binding sites on Gab1, downregulating RasGAP membrane recruitment and the ability to deactivate Ras [36]. Conversely, the SHP2 dephosphorylation of PI3K docking sites on Gab1 inhibits Gab1-PI3K association, downregulating PI3K/Akt signalling [38]. Thus, inhibiting the Grb2-Gab interaction should decrease Ras signalling but is unlikely to reduce PI3K/Akt signalling. Affimer C-C12 decreased phosphorylation of ERK but also reduced Akt phosphorylation, which conflicts with C-C12 inhibiting Grb2-Gab interactions.

Whilst our results are more aligned with C-C12 inhibiting SOS interaction rather than Gab, further work is required to confirm this. Structural insight into how and where the Affimer proteins bind the Grb2 SH3 domains would provide information concerning the mode of inhibition. For example, X-ray crystallography could be employed to determine the amino acid residues involved in the binding interactions and to detect if any structural changes in Grb2 occur when it is bound to Affimer. Also, through comparison with existing structures of Grb2 SH3 domains in complexes with ligands, it might be possible to predict if Affimer protein is likely to inhibit these interactions. As already discussed, allosteric effects may affect binding; therefore, it is preferable to study the interactions of full-length proteins.

Intriguingly, despite the differing outcomes in Grb2-facilitated cytoplasmic signalling, cells expressing either N-D7 or C-C12 resulted in elevated Grb2 levels in both serum-starved and stimulated cells. This suggests the SH3 domains may play a role in regulating Grb2 protein stability and are perhaps involved in targeting the protein for degradation. In eukaryotic cells, proteins are targeted for degradation via the proteosome (via ubiquitination) or via the lysosome [39]. Grb2 contains sites for ubiquitination within the NSH3 domain at Lys44 and Lys56 and within the SH2 domain at Lys109 (Figure 2b). If the binding of Affimer proteins to Grb2 blocks these ubiquitination sites, then this would explain these observed increases in Grb2 levels. However, none of these sites are within the CSH3 domain. Furthermore, it is unclear how these results fit with the model proposed in Figure 11. Therefore, further work is required to fully explore the potential involvement of Grb2 SH3 domains in regulating protein stability and the consequence of blocking this.

## 5. Conclusions

This study aimed to demonstrate the use of Affimer proteins as inhibitors of Grb2 SH3 domain interactions. We isolated and tested Affimer proteins targeting both the N- and C-terminal SH3 domains and found that targeting CSH3 alone appeared sufficient for downregulating Ras signalling in EGF-stimulated HEK-293 and U-2 OS cells. Whilst future work is required to determine what protein interaction partner Affimer C-C12 inhibits, this preliminary result does provide support for a previously proposed model in which the Grb2 CSH3 domain is necessary for the SOS1-based activation of Ras. Overall, the work presented here provides evidence that Affimer proteins can be used to probe the SH3 domain’s function within its endogenous setting.

## Figures and Tables

**Figure 1 biomolecules-14-01040-f001:**
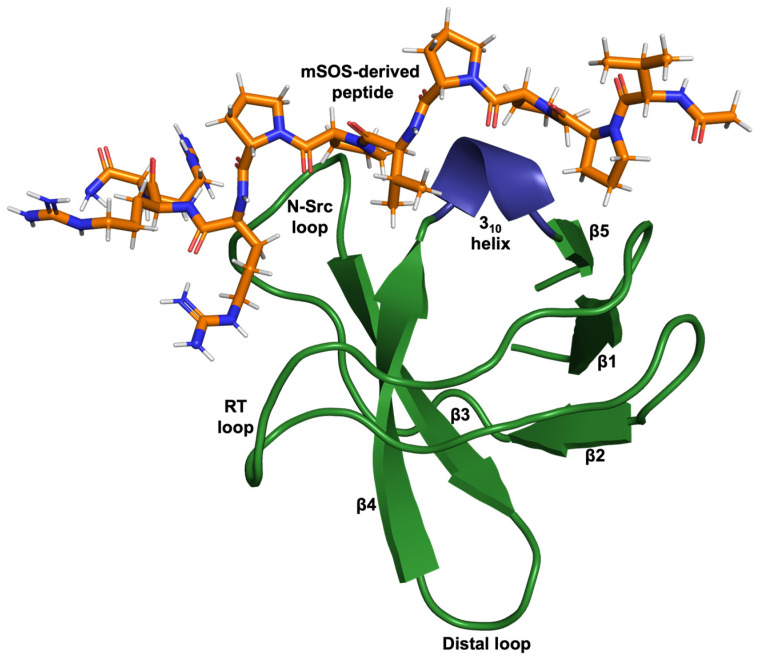
Cartoon-ribbon representation of the N-terminal SH3 domain of Grb2 complexed with a ten-residue ac-VPPPVPPRRR-nh2 peptide (stick model) derived from mouse SOS-1 sequence (PDB: 1GBQ) [2]. This image was generated using PyMOL.

**Figure 2 biomolecules-14-01040-f002:**
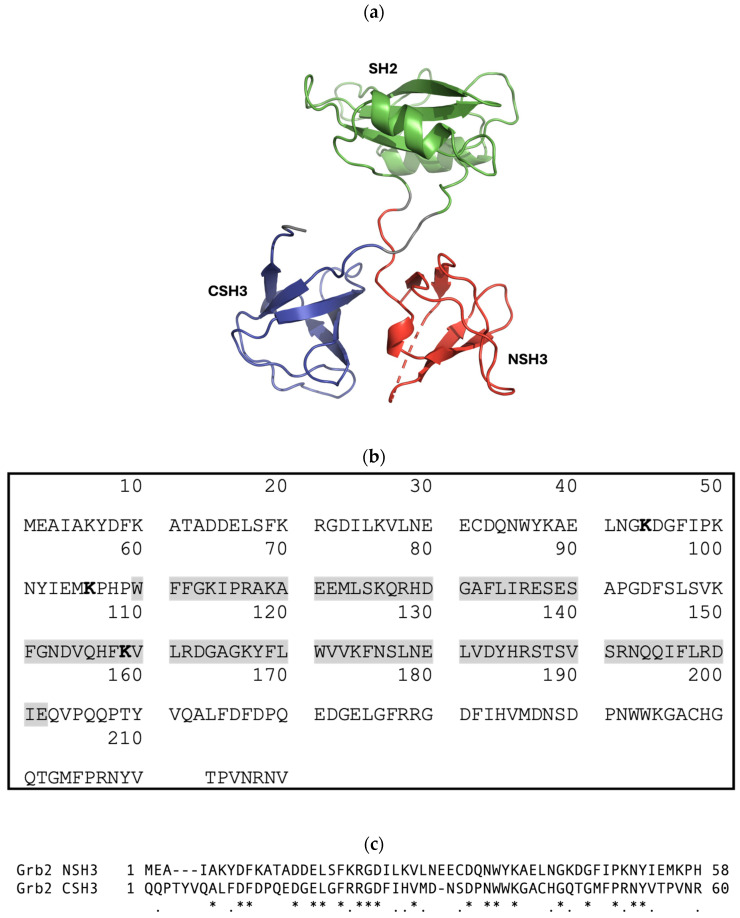
(**a**) Cartoon-ribbon representation of monomeric Grb2 depicting 3 distinct domains: N-terminal SH3 domain (NSH3) (red), SH2 domain (green), and C-terminal SH3 domain (CSH3) (blue). This image was generated using PyMOL from PDB: 1GRI [20]. (**b**) The canonical protein sequence of growth-factor-receptor-bound protein 2 (Grb2) from Homo sapiens. Grb2 (isoform 1) is a 217-amino-acid protein with a predicted molecular mass of 25,206 Daltons. Grb2 consists of an N-terminal SH3 domain (SH3N, residues 1–58), a central SH2 domain (residues 60–152, highlighted in grey), and a C-terminal SH3 domain (SH3C, residues 156–215). Sites for ubiquitination are highlighted in bold. Sources: UniProtKB-P62993 and https://www.genecards.org/cgi-bin/carddisp.pl?gene=GRB2 (accessed on 24 July 2024). (**c**) T-Coffee alignment (Myers-Miller algorithm) of the Grb2 NSH3 and CSH3 domains executed using MacVector 18.7.0. An asterisk (*) indicates identical residues; a period (.) indicates similar residues. The sequence identity is 32.8%, and the similarity is 52.5%.

**Figure 3 biomolecules-14-01040-f003:**
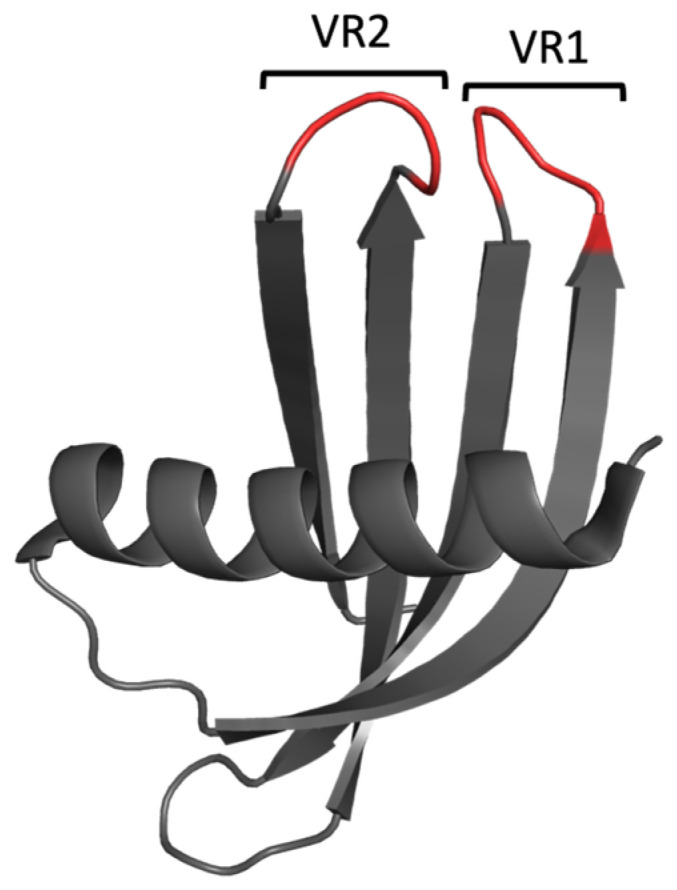
Cartoon-ribbon representation of the Adhiron/Affimer scaffold (PDB: 1NB5). The structural framework is depicted in grey, and the randomised regions for target selection (variable regions, VR) are depicted in red. This image was generated using PyMOL.

**Figure 4 biomolecules-14-01040-f004:**
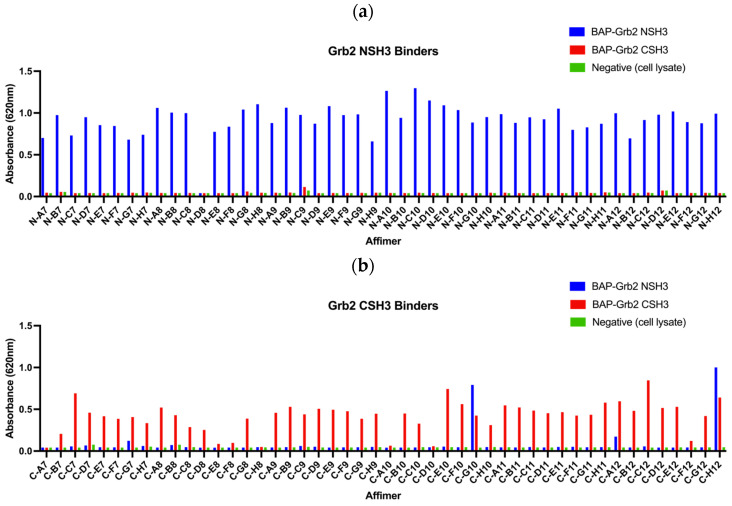
Phage ELISAs of 48 Affimer clones selected for binding to (**a**) biotinylated BAP-tagged Grb2 NSH3 domain and (**b**) biotinylated BAP-tagged Grb2 CSH3 domain.

**Figure 5 biomolecules-14-01040-f005:**
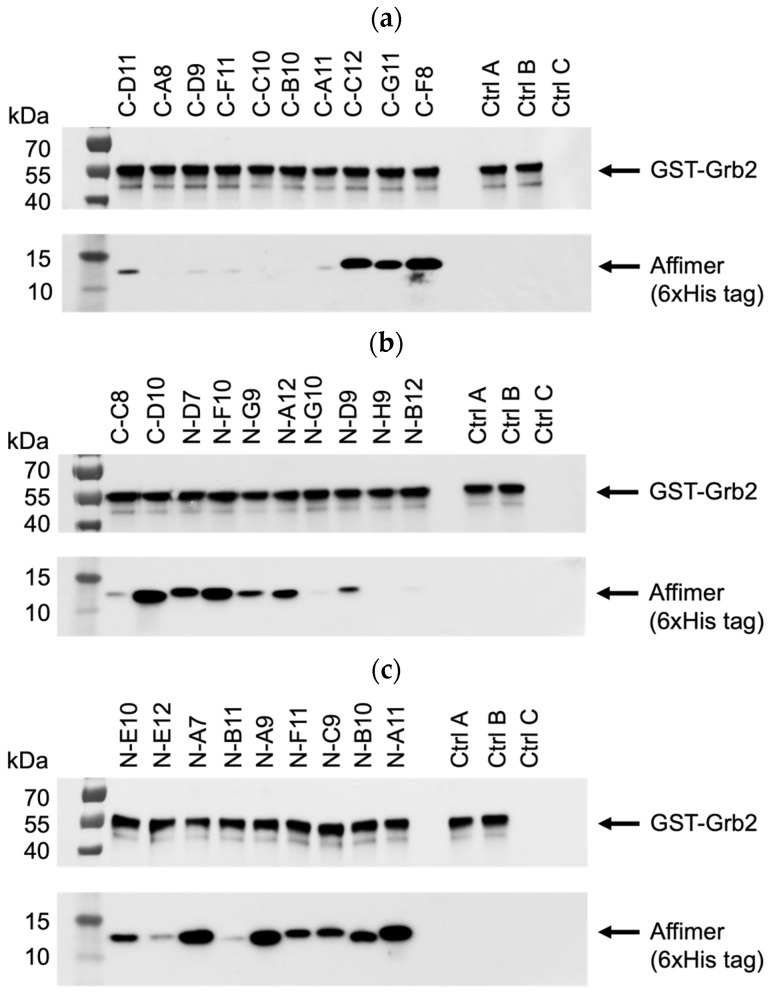
Representative anti-GST and anti-6xHis tag Western blots from three independent experiments (*n* = 3) demonstrating pull-down of interacting Affimer proteins with immobilised GST-tagged Grb2. Samples were divided over three SDS-PAGE gels (**a**–**c**) with control samples loaded on each: ctrl A was a non-Grb2 binding Affimer; ctrl B was a “no Affimer” control; and ctrl C was a “no GST-Grb2” control. See Appendix A for uncropped blots.

**Figure 6 biomolecules-14-01040-f006:**
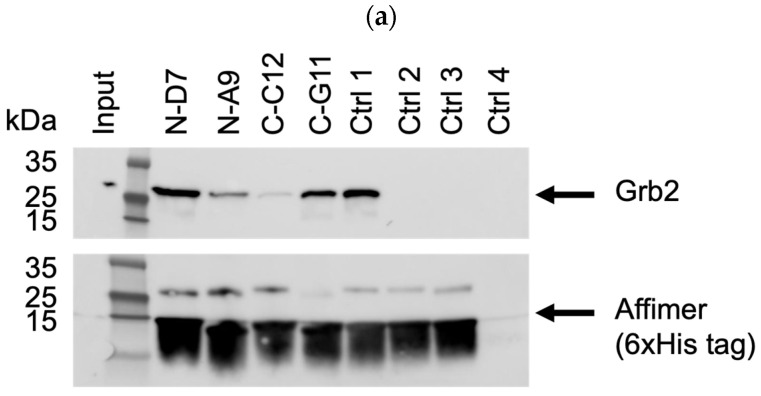
Representative anti-Grb2 and anti-6xHis tag Western blots from three independent experiments (*n* = 3) demonstrating pull-down of endogenous Grb2 from U-2 OS cell lysates with immobilized 6xHis-tagged Affimer proteins. Samples were divided over three SDS-PAGE gels (**a**–**c**) with control samples loaded on each: ctrl 1 was a binder of the Grb2 SH2 domain; ctrl 2 was a binder of the PLCG1 SH3 domain; control 3 was a non-Grb2 binding Affimer; ctrl 4 was a “no Affimer” control; ctrl 5 was another non-Grb2 binding Affimer; and ctrl 6 was an “Affimer only” (no cell lysate) control. See Appendix A for uncropped blots.

**Figure 7 biomolecules-14-01040-f007:**
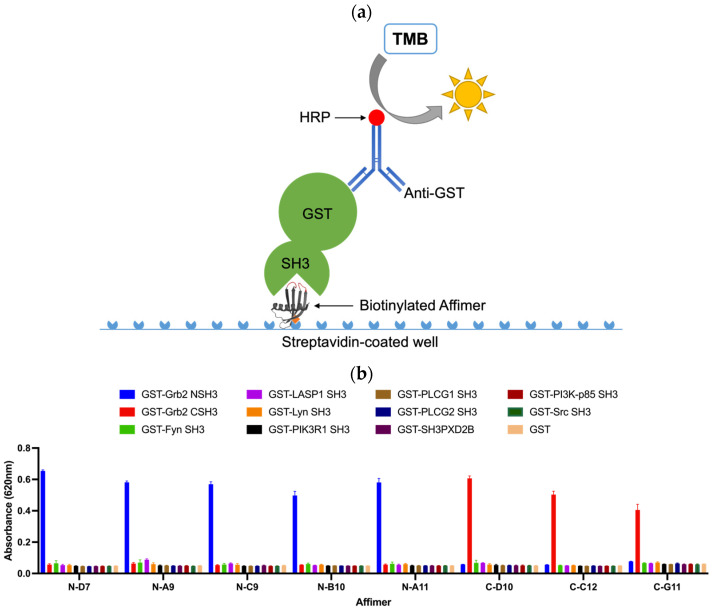
ELISA-based strategy for testing the cross-reactivity of Grb2-specific Affimer proteins. (**a**) Biotinylated Affimer proteins were immobilised in streptavidin-coated wells to capture interacting SH3 domains (GST fusions), which can be detected with an anti-GST (HRP) antibody. (**b**) ELISA test of the cross-reactivity of Grb2-specific Affimer proteins. Data are shown as the mean ± SEM of single analyses from three independent experiments (*n* = 3).

**Figure 8 biomolecules-14-01040-f008:**
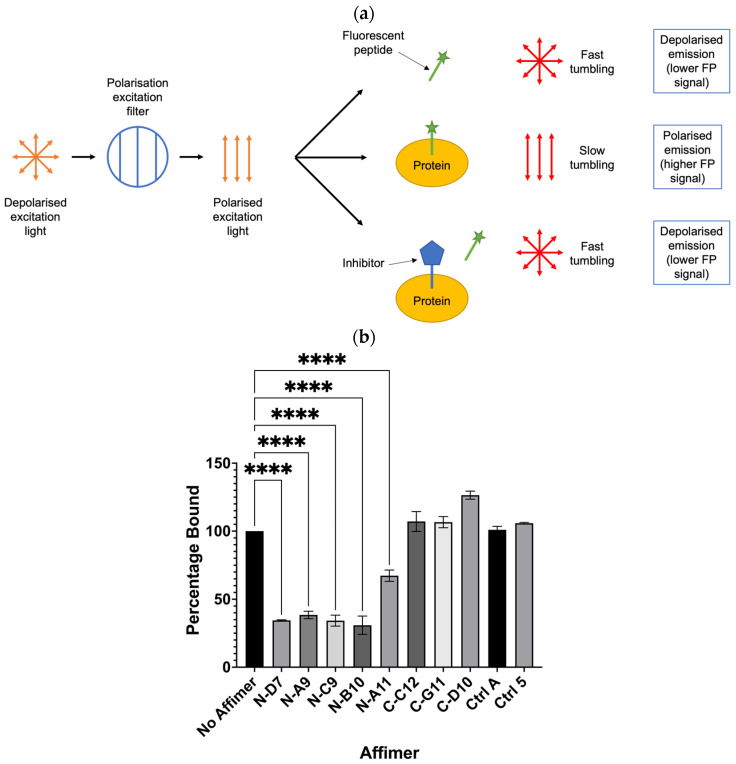
Fluorescence polarisation (FP) competition assay for testing the ability of Affimer to compete with FITC-labelled SOS-derived peptide for binding to full-length Grb2. (**a**) Schematic depicting the basic principle of an FP-based assay to determine if an inhibitor can displace binding of a fluorescent peptide from its target protein. (**b**) Affimer proteins targeting the Grb2 NSH3 domains significantly reduced binding of Grb2 with the SOS-derived peptide. FP values were normalised against FITC-peptide only and converted to percentage bound to Grb2 compared with the “No Affimer” control. Ctrl A and Ctrl 5 are the same controls described in Figure 5 and Figure 6. Graph shows mean ± SEM from three independent experiments (*n* = 3). A one-way ANOVA with Dunnett’s multiple comparisons test was applied, comparing the mean of each Affimer with the mean of the “No Affimer” control. **** *p* ≤ 0.0001.

**Figure 9 biomolecules-14-01040-f009:**
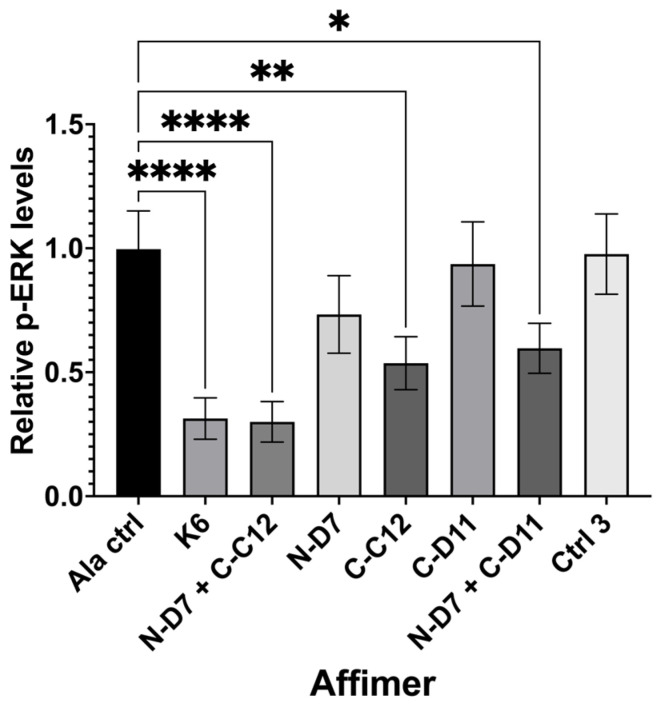
Quantification of nuclear phosphor-ERK (p-ERK) levels in transiently transfected HEK-293 cells producing turboGFP-tagged Affimer proteins. Graph shows nuclear p-ERK levels in Affimer-expressing HEK-293 cells relative to Ala control-expressing cells. K6 is a Ras-binding Affimer. Ctrl 3 is the same control described in Figure 6. Data represent mean ± SEM from three independent experiments (*n* = 3). A one-way ANOVA with Dunnett’s multiple comparisons test was applied, comparing the mean of each sample with the mean of the Ala control. * *p* ≤ 0.05, ** *p* ≤ 0.01, and **** *p* ≤ 0.0001.

**Figure 10 biomolecules-14-01040-f010:**
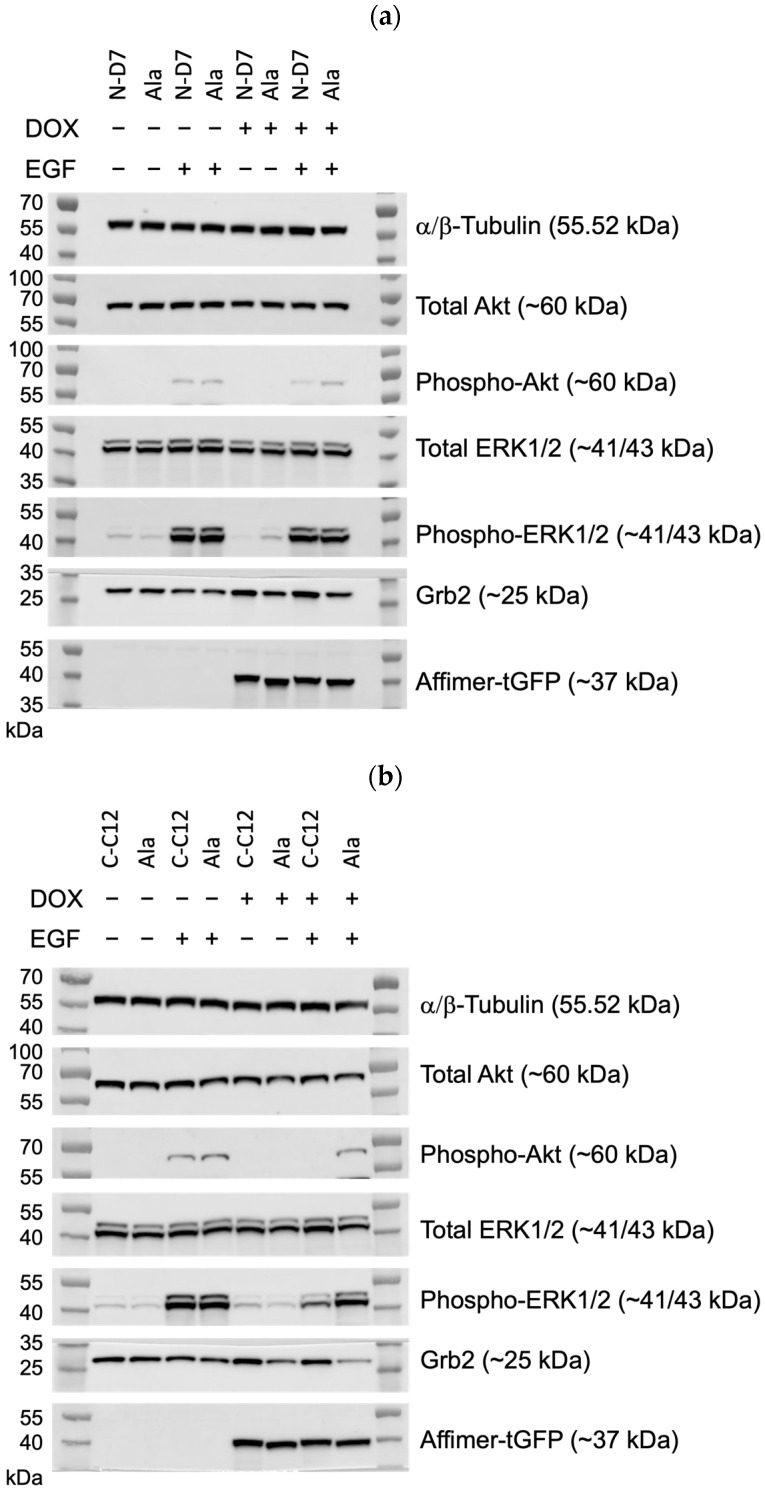
Epidermal growth factor receptor (EGFR) signalling assays conducted using U-2 OS cells stably producing turboGFP-tagged Affimer proteins. (**a**) Representative Western blots from three independent experiments of U-2 OS cells expressing Affimer N-D7 compared to Ala control (*n* = 3). See Appendix A for uncropped blots. (**b**) Representative Western blots from three independent experiments of U-2 OS cells expressing Affimer CC12 compared to Ala control (*n* = 3). See Appendix A for uncropped blots. (**c**–**e**) Quantification via densitometry of p-ERK/ERK, p-Akt/Akt, and Grb2 levels from Western blots as represented in (**a**,**b**). Signal intensities determined using ImageQuant™ TL were normalised to α/β-tubulin and quantified relative to Ala ctrl values. Statistical significance was determined using one-way ANOVA with Dunnett’s multiple comparisons tests. All datasets represent mean ± SEM from three independent experiments (*n* = 3). * *p* ≤ 0.05, ** *p* ≤ 0.01, *** *p* ≤ 0.001, and **** *p* ≤ 0.0001.

**Figure 11 biomolecules-14-01040-f011:**
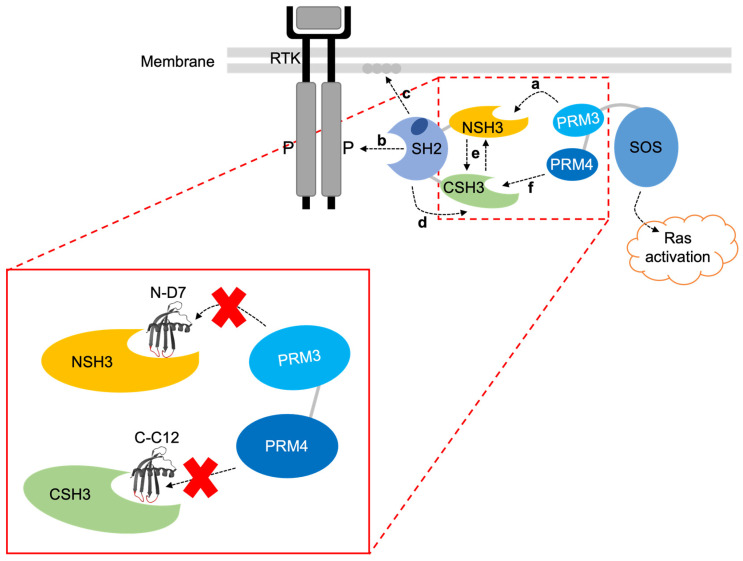
A proposed mechanism for the interaction of Grb2 with SOS1 [18,19]. The SH2 domain allosterically blocks the CSH3 domain, so only the NSH3 domain can bind SOS1. (a) SOS1 proline-rich motif 3 (PRM 3) interacts with the Grb2 NSH3 domain. (b) Cytosolic Grb2-SOS1 complex translocates to the tyrosine-phosphorylated RTK where the SH2 domain binds. (c) Grb2 SH2 domain also interacts with phospholipids in the plasma membrane. (d) This releases the block on the CSH3 domain, (e) triggering cooperative interactions between the two SH3 domains. (f) This enables the CSH3 domain to interact with SOS1 proline-rich motif 4 (PRM 4), inducing activation of Ras. The image in the red box shows where Affimer N-D7 and C-C12 are predicted to bind and inhibit interactions with SOS1 PRM 3 and PRM 4, respectively. Note: SOS1 PRM 3’s sequence is EVPVPPPVPPRRRPESAPAESSPSKI, and SOS1 PRM 4’s sequence is LDSPPAIPPRQPTSK.

**Table 1 biomolecules-14-01040-t001:** Amino acid sequences for Affimer clones selected for binding to Grb2 NSH3 and CSH3 domains. Only the variable regions (VR1 and VR2) are shown. Binders with AAE sequence in VR2 were selected from an Affimer library in which residues in VR1 only were randomised. Proline residues are highlighted in bold, and sequences that match known SH3-domain-binding consensus sequences are underlined. (Note: Frequencies do not total 48 (100%) as some clones did not produce viable sequences.).

Affimer Name	VR1 Sequence	VR2 Sequence	Frequency	Matching Clones
N-D7	P IMVGEWNV	HRWFHKRFQ	10/48 (20.8%)	N-B8, N-C8, N-H8, N-B9, N-E9, N-A10, N-C10, N-D10, N-E11
N-F10	SVHKTDRFL	WKMIVTPPP	9/48 (18.8%)	N-G7, N-A8, N-F8, N-H10, N-C11, N-D11, N-H11, N-H12
N-A9	WYPSIIKPY	QTQFNNIPP	7/48 (14.6%)	N-F7, N-H7, N-G11, N-C12, N-F12, N-G12
N-F11	HRVGKHIMI	FNFSEPQQP	2/48 (4.2%)	N-C7
N-G9	IHVPPFDTW	YHRGSSWRI	2/48 (4.2%)	N-F9
N-A12	SDMYPKLHA	HISYGIAAN	1/48 (2.1%)	
N-C9	SFVSKWKPY	DMPMSTKLK	1/48 (2.1%)	
N-B10	WAHHMEPVA	FMGEGVFWI	1/48 (2.1%)	
N-G10	WVQWQFEDM	YPGDMAYVA	1/48 (2.1%)	
N-D9	HTFGRNVTE	FEWMWNPAN	1/48 (2.1%)	
N-H9	HPYQVPVPA	HRRPKYIVP	1/48 (2.1%)	
N-B12	KWDDLAWWP	ADFMLWEFV	1/48 (2.1%)	
N-A11	NPYSPTVSG	YIYPKPTKY	1/48 (2.1%)	
N-E10	P EQQAYQYN	YFAPSTWRI	1/48 (2.1%)	
N-A7	EPFVPRTAWWW	QERQNHNMM	1/48 (2.1%)	
N-E12	DGSPEKQI	GGRPWFIGR	2/48 (4.2%)	N-E7
N-B11	MIPHVE	AVWQRINFE	2/48 (4.2%)	N-E8
C-D11	IMRPYWASS	VGGDAYEKM	1/48 (2.1%)	
C-A8	VMRPWWDSS	AAE	16/48 (33.3%)	C-E7, C-F7, C-D8, C-G8, C-A9, C-B9, C-E9, C-F9, C-G9, C-H10, C-B11, C-C11, C-E11, C-H11, C-B12
C-D9	IMRPWWHSS	AAE	1/48 (2.1%)	
C-F11	IMRPWYMSS	AAE	1/48 (2.1%)	
C-C10	IQRPWFYSS	AAE	1/48 (2.1%)	
C-B10	IQRPFYDSS	AAE	2/48 (4.2%)	C-C9
C-A11	IKRPWWASS	AAE	1/48 (2.1%)	
C-C12	RNIPIYP P Q	FEKPTSMHH	3/48 (6.3%)	C-C7, C-E10
C-G11	GPWGSVYMW	RETGNPTWL	5/48 (10.4%)	C-B7, C-H8, C-A10, C-F12
C-F8	DPYVIEVKD	AILGLIQPQ	1/48 (2.1%)	
C-C8	AVQHWYPVQ	HGGHYRAPM	1/48 (2.1%)	
C-D10	GNEIMHYNV	SWPTYWEPN	1/48 (2.1%)	

## Data Availability

The original contributions presented in the study are included in the article/Appendix A, further inquiries can be directed to the corresponding author/s.

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
