# Peer review of "Targeting Grb2 SH3 Domains with Affimer Proteins Provides Novel Insights into Ras Signalling Modulation"

_biomolecules, 2024, doi:10.3390/biom14081040_

Round 1

Reviewer 1 Report

Comments and Suggestions for Authors

The article entitled “Targeting Grb2 SH3 domain with affimer proteins reveals novel insights into Ras signalling modulation” reports the isolation of Affimers that selectively bind to the N- and C-terminal SH3 domains of Grb2. Binding to targets was tested with multiple ways, using both full-length Grb2 and isolated domains. Affimers targeting the NSH3 domain were shown to block interaction with the SOS1 peptide in the purified condition. Finally, selected NSH3 and CSH3-specific Affimers were expressed in cells to determine if they inhibit Ras signalling. They found that targeting CSH3 alone could inhibit EGF-stimulated Ras-MAPK signalling. Overall, this is an important work that obtained useful Affimers for the study of signal transduction, but more background needs to be provided for non-experts to follow the discussion.

1. A brief overview of the Affimer scaffold, including the positions of VR1 and VR2, is needed in the introduction. It would also be helpful to show the domain organization of Grb2 and the sequence identity between the N-SH3 and C-SH3 domains in Figure 1.

2. Have competitive assays tested whether Affimers targeting CSH3 can inhibit the interaction with SOS1-PRM4 in purified conditions?

3. The discussion in lines 545 to 558 is confusing.

4. Minor points. 

Line 27, “found in” is duplicated.

Lines 381, 383, and 413 “N D7” and “C D10” should be “N-D7” and “C-D10”.

Figure 9b. SH2" should be indicated in the figure.

Reviewer 2 Report

Comments and Suggestions for Authors

In this manuscript, Tang et al describe the use of affimer proteins to inhibit the growth factor receptor-bound protein 2 (Grb2), particularly focusing on the blocking of the interaction with a protein domain of Son of sevenless (SOS), the guanine nucleotide exchange factor (GEF) of Ras, a signaling cascade. The study begins with the development of affimer variants against the N-term and C-term SH3 domains of Grb2 through phage display. Interestingly, not all the positive obtained variants contained the PXXP motif in their sequences. Afterwards, the authors check that their affimer variants are specific towards their targets, using an ELISA assay, and by fluorescence polarization competition assays they confirm the inhibitory effect with respect a SOS-derived peptide. Finally, the authors see, through experiments with HEK-293 and U-2 OS cells, that an inhibitory effect in Ras signaling pathway is produced mostly by the use of their C-term-targeted affimer variants, thus blocking the C-terminal SH3 domain of Grb2. The experiments are finely designed, and results are coherent and sound. The manuscript is well written and can be easily followed. Thus, I have no complaint or doubt in accepting the manuscript in its current form. I just want to add a commentary of encouraging the authors to obtain the crystal structures of the described complexes and/or studying the Kd of the interactions. But those experiments may be a follow-up; in my opinion, the current paper is itself completed.
